# Preservation of microscopic fur, feather, and bast fibers in the Mesolithic ochre grave of Majoonsuo, Eastern Finland

Tuija Kirkinen[1]*, Olalla López-Costas[2,3], Antonio Martínez Cortizas[4], Sanna P. Sihvo[5,6], Hanna Ruhanen[5,6], Reijo Käkelä[5,6], Jan-Erik Nyman[7], Esa Mikkola[7], Janne Rantanen[7], Esa Hertell[8], Marja Ahola[1], Johanna Roiha[1], Kristiina Mannermaa[1]

**1** Archaeology, Department of Cultures, University of Helsinki, Helsinki, Finland, **2** Area of Archaeology, Department of History, EcoPast (GI-1553), CRETUS, Universidade de Santiago de Compostela, Santiago, Spain, **3** Archaeological Research Laboratory, Wallenberglaboratoriet, Stockholm University, Stockholm, Sweden, **4** CRETUS, EcoPast (GI-1553), Fa4culty of Biology, Universidade de Santiago de Compostela, Santiago, Spain, **5** Helsinki University Lipidomics Unit (HiLIPID), Helsinki Institute of Life Science (HiLIFE) and Biocenter Finland, Helsinki, Finland, **6** Faculty of Biological and Environmental Sciences, Molecular and Integrative Biosciences Research Programme, University of Helsinki, Helsinki, Finland, **7** Finnish Heritage Agency, Helsinki, Finland, **8** Museums of Lappeenranta, Lappeenranta, Finland

* tuija.kirkinen@helsinki.fi

**Data Availability Statement:** The excavation report and find catalogues are available in Finnish Heritage Agency public database in https://asiat.

## Abstract

The study of animal and plant fibers related to grave furnishing, garments, and grave goods in thousands-of-years-old burials provides new insights into these funerary practices. Their preservation presupposes favorable conditions, where bacterial and fungal activity is at a minimum, as in anaerobic, wet, salty, arid, or frozen environments. The extreme acidic-soil environments (i.e., podzols) of Finland pose a challenge when it comes to studying funerary deposits, as human remains are rarely found. However, its potential to preserve microparticles allows us to approach the funerary event from a totally different point of view. Here, we present the first multiproxy analyses of a Mesolithic deposit from Finland. A red-ochre burial of a child found in Majoonsuo is studied by analyzing 1) microscopic fibers, 2) fatty acids, and 3) physical-chemical (CIELab color, pH, grain size) properties of 60 soil samples and associated materials. The microscopic fibers evidenced the remains of waterfowl downy feathers, a falcon feather fragment, canid and small rodent hairs as well as bast fibers. These could have been used in furnishing the grave and as ornaments or clothes. Canid hairs could belong to a dog inhumation, or more likely to canid fur used as grave good/ clothes. Samples with microparticles have more long-chain and unsaturated fatty acids, although animal species identification was not possible. Soil properties indicate that the burial was made in the local soil, adding homogeneous red ochre and removing the coarser material; no bioturbation was found. The highly acidic sandy soil, together with a slight increase in finer particles when ochre is abundant, probably resulted in micro-scale, anoxic conditions that prevented bacterial attack. This study reveals the first animal hairs and feathers from a Finnish Mesolithic funerary context, and provides clues about how their preservation was possible.

museovirasto.fi/case/MV/80/05.04.01.00/2020. All
other relevant data are within the paper and its
Supporting Information files.

**Funding:** - KM: Animals Make Identities (AMI),
ERC Consolidator Grant No 864358, https://erc.
europa.eu/ - OLC: Xunta de Galicia through Grupos
de Referencia Competitiva (ED431C 2021/32),
Ramon y Cajal Fellowship (RYC2020-030531-I),
https://www.deusto.es/cs/Satellite/deusto/en/
university-deusto/admissions-administration-and-
grants/scholarships-and-grants-/ramon-y-cajal-2/
beca - OLC: Plan Nacional de investigation Pollutio
(PID2019-111683RJ-I00) from Ministerio de
Ciencia e Innovación de España, https://www.
ciencia.gob.es/ The funders had no role in study
design, data collection and analysis, decision to
publish, or preparation of the manuscript.

**Competing interests:** The authors have declared
that no competing interests exist.

## Introduction

Cloth-type [1] materials made of organic soft tissues have been crucial throughout the Prehistoric Era in producing garments, containers, tents, ropes, strings, nets, and blankets [2–5]. In burial rituals, they served in wrapping and covering the dead, in furnishing the grave, for clothes and accessories, and for grave goods [6–8]. However, our knowledge about the Stone Age use of fibers and skins is rather limited, as organic materials decompose easily in soils. The survival of fur, hairs and feathers, plant fibers and skin implies favorable conditions in anaerobic, wet, salty, arid, or frozen environments where bacterial and fungal activity is at its minimum [9–12]. Even if the fibers themselves have decayed, their imprints e.g. in burnt clay carry information of hairs and plant fibers [13]. From the Bronze Age onwards, the proximity to metal artifacts has preserved organic materials due to metal alloys that have inhibited the decaying of the material [14–16].

Soil pH is important for the preservation of fibers as well as for human remains. In acidic environments, the survival of keratin fibers (animal hair and feathers) is favored, while cellulose-based materials (e.g., bast fibers) degrade rapidly [12,17]. In Northern Europe, human remains and unburnt animal bones are rarely recovered in Stone Age contexts [18–20] due to the acidity of podzol soils. Low soil pH, indicating higher [H+], interact with the mineral fraction of bone and enamel (i.e. bioapatite) altering them. pH has been studied as a key factor for bone preservation (see a summary in [21]). In contrast to the weathering of the mineral component of the skeletons, acidic conditions usually limit the microbial attack on organic remains, enhancing their preservation. However, as important as the acidity of a soil is, other pedological conditions, such as grain size, can also determine the degree of preservation. Sandy soils are more likely to be drained and water would percolate deep in the soil, preventing anaerobic conditions and facilitating microbial alteration; meanwhile, soils which have higher silt and clay fraction content are more prone to being water-saturated (i.e., anaerobic conditions) promoting the preservation of fibers and other organic materials. It is not clear how red ochre could affect the preservation of organic and inorganic remains. On the one hand, the burials which have additional red ochre could have increased the fine particle fraction (silt and clay) and may promote microscale anaerobic conditions. This being the case, it would also help to preserve fibers embedded in ochre. On the other hand, red ochre normally contains a large proportion of iron oxides that can also affect organic matter preservation; additionally, iron limits the access of bacteria to nutrients such as phosphorous [22]. It is expected that red ocher causes changes at the micro-scale, creating a map of microenvironments in accord with their abundance. Until now, most studies on red ocher focused specifically on the ocher "nature" (see a review in [23]), ignoring the fact that the physical-chemical conditions of the parent soil will condition the processes that dominate both inside and outside the ocher area. A pedological analyses of soils with different amounts of red ochre would be necessary to shed light on the dominant processes (preservation or degradation), which are associated with red ocher in a specific burial.

Taking this all into consideration, we hypothesized that, in Finland, the acidity of soils (pH 4–5) might have prevented the decaying of keratinous fibers even in Stone Age contexts. Our focus is on microscopic fibers, by which we refer to 100-1000-μm-long animal hairs, feathers and plant fibers, termed in the research literature as microparticles [24], microfossils, microresidues or, in sediment samples, as non-pollen palynomorphs (NPP, [25]). These kinds of fibers have been found from dental calculus [26,27], coprolites [28], on the surface of artifacts [29–31], and from pollen and NPP analyses made from the sediments of archaeological sites [32–37]. Fibers and feathers have also been the main subject of research in analyzing soil samples taken from burials [19,38–40].

Therefore, we conducted a large-scale soil sampling in a Stone Age inhumation burial in Majoonsuo, eastern Finland, excavated in 2018 by the Finnish Heritage Agency. The investigated burial form is designated in Finland as a "red-ochre grave" (Fi. *punamultahauta*) in the sense of a Stone Age burial where an ochre feature indicates a burial, but other finds are scarce or absent [41]. Here, our aim was to detect microscopic evidence of the use of cloth-type materials in a child burial in which only tooth enamel and four quartz artifacts were recovered. In this paper, we present the results of microparticles and chemical soil analysis of the Majoonsuo grave. Our investigation had two goals; first, to detect microscopic fiber remains from the soil samples and, second, to analyze the factors that have potentially affected the preservation of fibers in Majoonsuo. Finally, the significance and potential of micro-remains for the study of mortuary practices is discussed.

## Site and field work

The Majoonsuo ochre grave is located in the municipality of Outokumpu, Eastern Finland. It is one of the many prehistoric sites of the area. The burial was discovered in 1992, when ochre from a possible Stone Age inhumation grave was spotted on the surface of a recently dug service trail. The trail runs on the western slopes of a small and wooded moraine hill currently surrounded by drained bogs and a small lake Fig 1.

In 2018 the Finnish Heritage Agency conducted a rescue excavation at the site, since the grave was considered to be endangered by erosion, forestry, and occasional motor traffic. An area of 24 m$^2$ was opened along the service trail, from which an approximately ca. 10–70-cm-thick layer of gravelly sand had been removed to create the trail (Fig 2). The site was excavated in successive layers of 2.5 cm, and the removed material was sieved. The finds and features were documented by GNSS application Topcon Hiper SR, the accuracy of which was ±2 cm. The excavation layers were documented by drawing and photographing. The find material is archived in the National Museum of Finland (KM 41572:1–211), and the detailed excavation report [42] is available at https://asiat.museovirasto.fi/case/MV/80/05.04.01.00/2020.

## Features and find material

### The grave feature

The grave feature measured 1.0 x 0.4 meters and was rectangular-shaped with rounded corners. Its orientation was approximately northwest-southeast. The ochre-stained sand continued to a depth of 62–64 cm below the surface of the trail. Above the grave, approximately 10–20 cm of sand—the filling layer of the grave—had been removed during the construction work. We assume that an excavator bucket tooth had also cut through the grave, tearing the upper part of it in half and forming a 1.5-m-long stripe of ochre on the surface of the trail. However, the rest of the grave appeared to have remained intact. See S1 Appendix.

**Human teeth.** Due to the acidity of the soil, only seven human tooth fragments were preserved from the deceased. The teeth remains were found in a soil sample and in sieved material, originating at a depth of 5.0–7.5 cm (layer 3) below the trail surface. Most probably, the teeth came from the same context.

All tooth fragments are unburnt and covered by a layer of red ochre. They have only the enamel part, no roots have survived. The teeth were identified with the naked eye based on the morphology and without cleaning the soil away from the surface. As a result, we suggest that the teeth belong to one individual of an age between 3.5 and 10.5 years, but further analyses must be conducted to estimate a more precise age Fig 3.

**Quartz artifacts.** Eight pieces of red ochre-stained quartz objects were found in the grave at a depth of 2.5–12.5 cm. Four of these pieces have been identified as tools: two as transverse

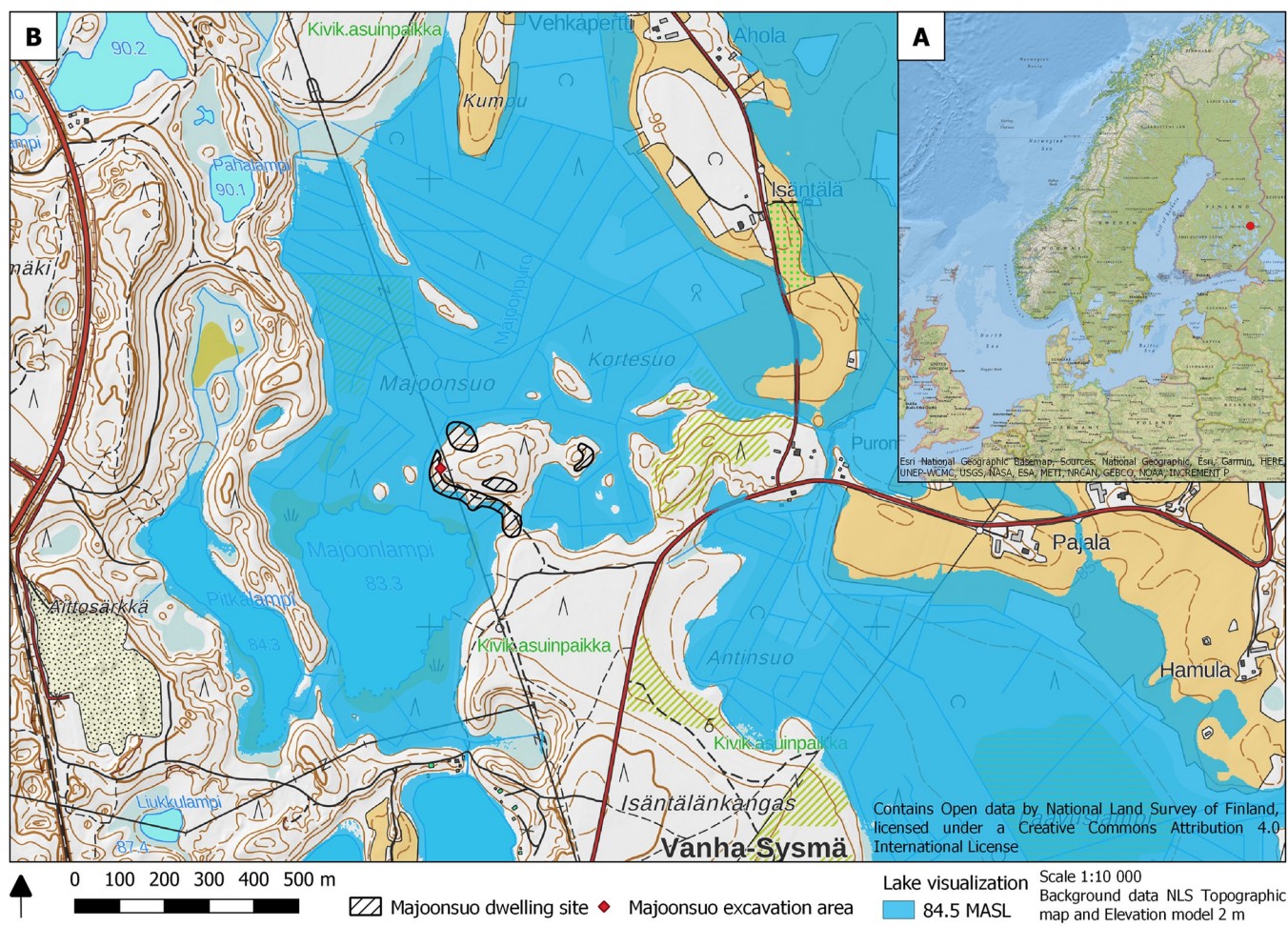

**Fig 1. A) The location of Majoonsuo in Finland.** Background map: ESRI National Geographic Basemap, sources National Geographic, ESRI, Garmin, HERE, UNEP-WCMC, USGS, NASA, ESA, METI, NRCAN, GEBCO, NOAA, INCREMENT P. CCBY 4.0 International License. **B) Majoonsuo dwelling site and excavation area.** The Mesolithic water level visualization is 84.5 meters above sea level (MASL). The site has been located on a sheltered lakeshore on a small peninsula. Background map: National Land Survey of Finland, CCBY 4.0 International License. The geographical information of archaeological sites is based on the Finnish Heritage Agency Register, CCBY 4.0 International License. Drawing: Johanna Roiha.

arrowheads (KM 41572:45 and:98) and two as small, retouched items (:101 and:102). Another ochre-stained retouched quartz tool (:48) was found just north of the grave and may have originally been situated inside the grave, but later dislocated during the construction of the trail. No more artifacts were found deeper in the soil even though the ochre-stained sand continued to a depth of 62–64 cm below the surface of the trail Fig 3.

**Ochre.** The grave was heavily stained with ochre, a practice commonly documented in hunter-gatherer burials from the Palaeolithic period onwards [43]. The term ochre refers to iron precipitates ($Fe_2O_3$), formed naturally in mires, springs, lakes, and podzol soils [44,45]. Its beautiful red color can be made even more intense by burning [46,47] through production of hematite, creating an intense red color at low temperature [48].

*Stones.* Plenty of ochre-colored natural stones were found in the grave area. Although the area surrounding the grave was naturally marked by rocky ground, some of these stones were heavily ochre-stained and could thus have been positioned near the grave on purpose [49].

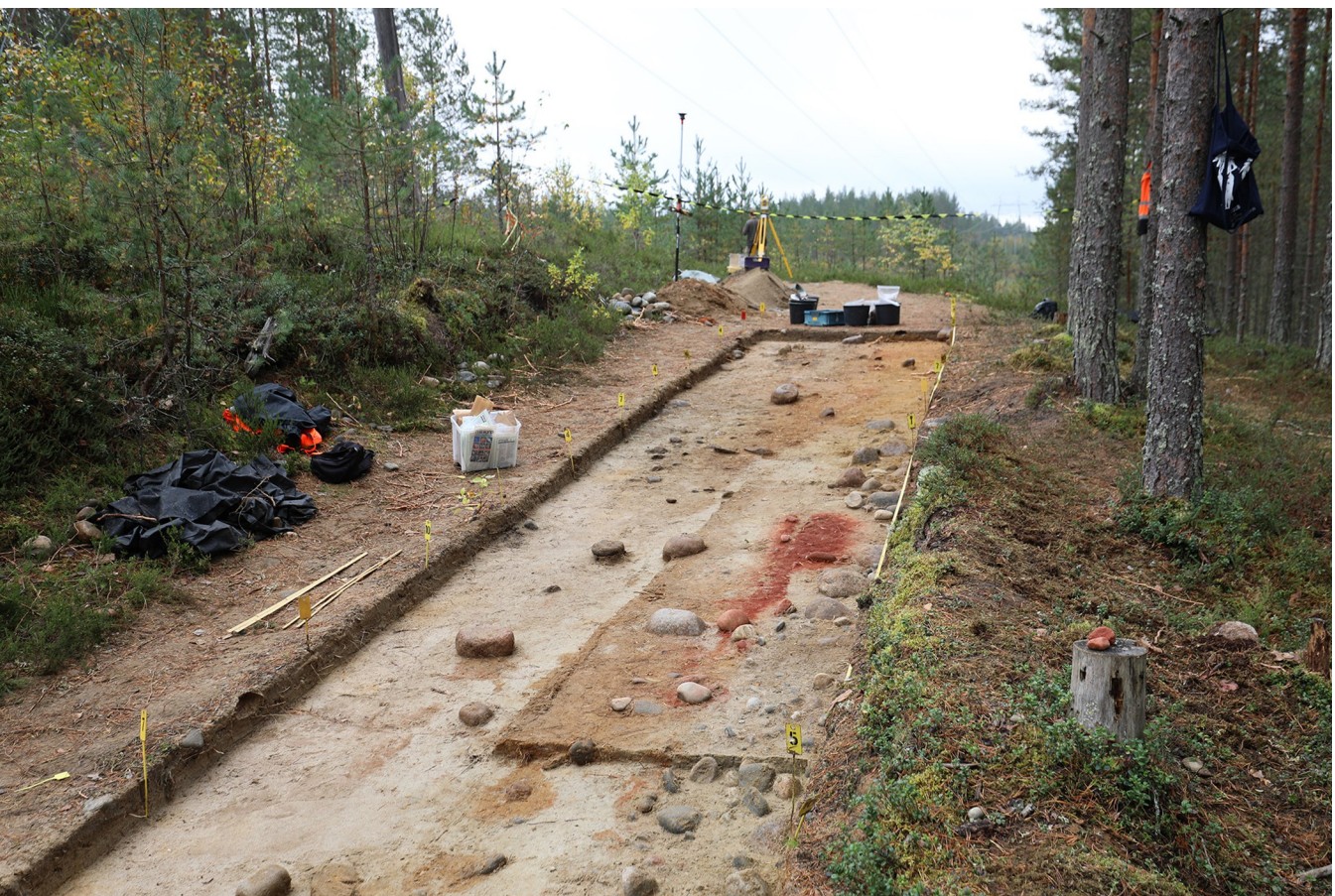

**Fig 2. Excavation area towards SE.** Photo: Kristiina Mannermaa.

## The habitation area

On the southern side of the grave, a 20-cm-thick cultural layer consisting of discolored and partially burned sand was found under the disturbed surface soil. The settlement-site finds comprise predominantly quartz flakes, quartz artifacts and burned bone. Most bones could not be identified within any taxon. Two fragments belong to either red fox (*Vulpes vulpes*) or dog (*Canis familiaris*), and one fragment to an unspecified seal (Phocidae). Fish are represented by perch (*Perca fluviatilis*), pike (*Esox lucius*) and some cyprinid (Cyprinidae) species. One mammal bone fragment is probably from a grooved artifact [50].

## Dating

The research area is located 89.9 meters asl. Relatively soon after the end of the last Glaciation, the site emerged from the waters, but was covered again because of the but was covered again because of the transgression of Ancient Lake Saimaa during the Late Mesolithic (c. 6800–5200 cal BC; the maximum shore level in the region is 92 m asl). After the breakthrough of the River Vuoksi from Lake Saimaa in the early 4th millennium BC, the water level of the lake sank rapidly [51–54]. According to shore-level dating, the possible time frames can be found in the Early Mesolithic Stone Age and the Typical Comb Ware phase (3800–3400 calBC).

A small number of burned bone fragments collected from the settlement layer was $C^{14}$-dated to 8354±37 BP (Ua-64385). The calibration with OxCal v.4.3.2 IntCal13, the 68.2%

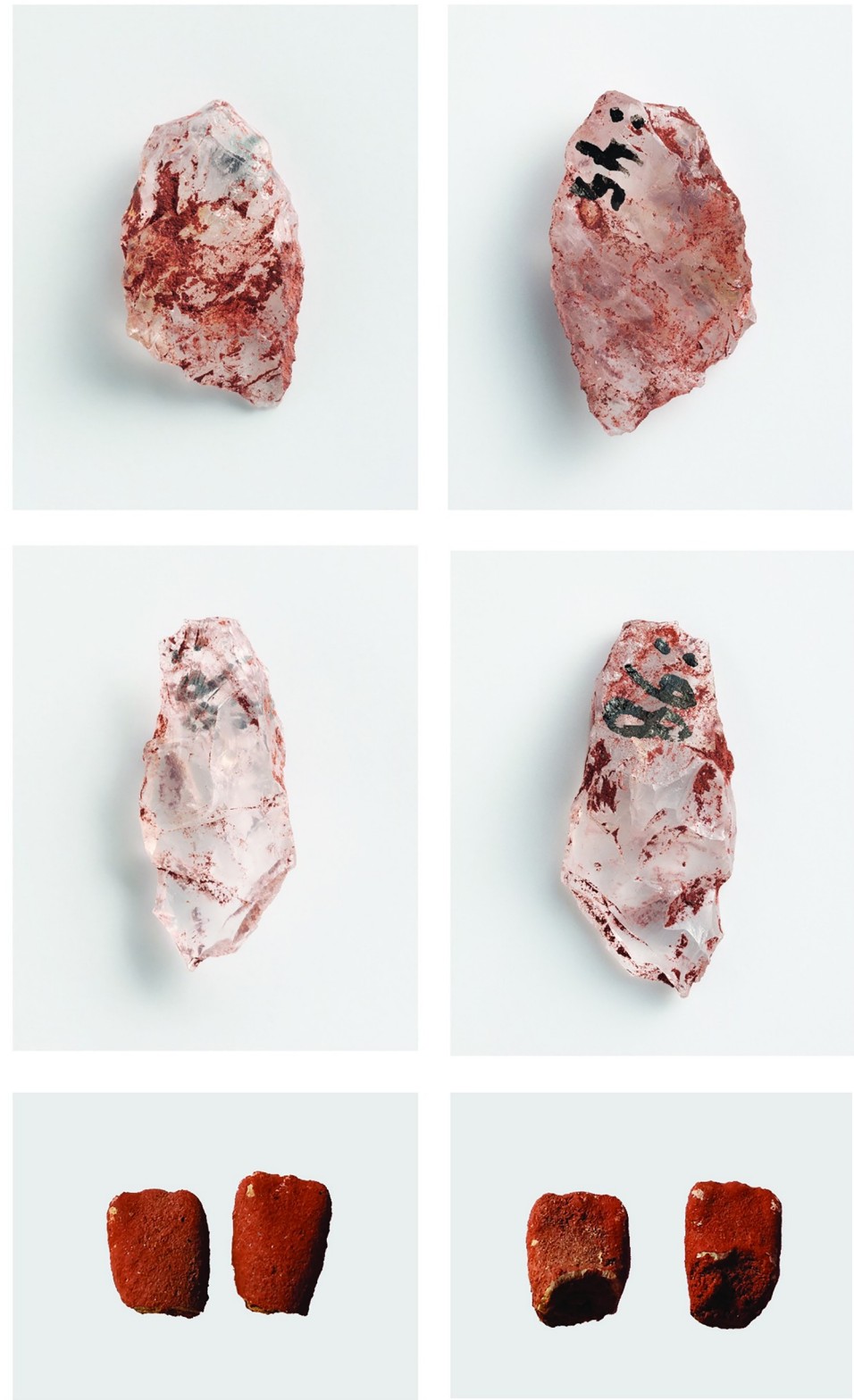

**Fig 3. The find material of the grave: Transverse arrowheads abobe: 13x8x3 mm (KM 41572:45), and middle: 15x7x3 mm (:98).** Below: Two permanent mandibular incisors (KM 41572:47); Photos Ilari Järvinen / Finnish Heritage Agency, Archaeological collections and Kristiina Mannermaa.

probability level falls between 7494–7447, 7432–7426 or 7413–7360 calBC and with 95.4% probability level between 7521–7337 calBC dating the dwelling site to the Mesolithic Stone Age [42]. This approximation was supported by the lack of any ceramic find material. The grave itself did not include anything that could be radiocarbon dated. The transverse arrowheads found in the grave date to the Late Mesolithic Stone Age; however, the radiocarbon dating of the site seems anomalously old given the currently known temporal distribution of the Late Mesolithic arrowheads [55,56]. This seems to indicate more than one use period of the site during the Mesolithic. To conclude, the Majoonsuo grave and the nearby settlement site survived regardless of the transgression of Ancient Lake Saimaa. After the regression, the southern slopes of the hill were populated again by the Neolithic Asbestos Ware groups.

## Materials and methods

### Collection of samples

An extensive collection of 60 soil samples of 620–3,388 g were taken during the excavation (S1 Appendix). The permits for archaeological excavation, analyzing the soil samples, and studying the find material were obtained by the Finnish Heritage Agency. All necessary permits were obtained for the described study, which complied with all relevant regulations.

The samples covered the most intensive ochre concentration from top to bottom as well as the secondarily colored areas. One reference sample was taken outside the grave formation. Soil samples were packed in plastic zip bags. The samples from layer 3 were kept in a fridge and the others were stored at room temperature.

The samples were delivered to three laboratories, the Archaeological laboratory of the University of Helsinki for microparticle analysis, the Lipidomics unit (HiLIPID) of the University of Helsinki, Finland for identification of fatty acids (FA), and EcoPast laboratories, University of Santiago de Compostela, Spain for soil/sediment analysis.

### Microparticle analysis

In total, 23 soil samples from the grave, from the secondarily colored areas and a reference sample taken outside the grave, were analyzed for microfibers in the Archaeological laboratory of the University of Helsinki. The samples were prepared in the soil laboratory, and the microscopy was conducted in the microscope room. The contamination of samples by modern fibers was prevented by intensive cleaning of the surfaces and by taking control samples from the table and other surfaces by tape.

From each bag, a subsample of 150–300 g was separated. Because of the intensive red color caused by ochre, which prevented the visibility of fibers, the soil was water sieved gently with a 0.125 mm sieve following [57]. The extracted material was stored in 15 ml conical centrifuge tubes. The tubes were centrifuged for 7 min at 2500 rpm by the TD4A-WS desk centrifuge.

The samples were prepared for transmitted light microscope examination by pipetting the material on microscope slides and by covering them with coverslips. The material was studied with visible and polarized light microscopy, using a Leica DM 2000 LED microscope and Amscope 40X-1600X Advanced Professional Biological Research Kohler Compound Microscope with 100x - 400x magnification. The material was documented with Leica ICC50 W and 10MP USB3.0 cameras. A selection of fibers was prepared for scanning electron microscopy (SEM) imaging by placing them on double-sided carbon tape that had been fixed on aluminum stubs. The SEM samples were coated with a Leica ACE 600 sputter coater with a 10 nm layer of Au-Pd. SEM imaging was performed with Zeiss Sigma VP and using an acceleration voltage of 3 kilovolts.

For the identification, the keys on [10,58,59] were applied, and the fibers were compared with a reference collection of Fennoscandian species, collected at the Natural History Museum, University of Helsinki. Feathers were identified by their morphology after [60]. The terminology followed mainly [59] and [60].

## Soil/Sediment analysis

Sub-samples of 100g of fine earth (<2 mm) were analyzed for physical (color, grain size) and chemical (pH) properties. Color was quantified using CIELab color space that provides five parameters: luminosity (L, black is 0 and white is 100), the green (negative values) to red (positive values) component (a*), the blue (negative values) to yellow (positive values) component (b*), chromaticity (C*) and hue (h). Quantitative color was determined in finely milled (<50 μm) samples with a Konica-Minolta CR5 colorimeter for solids located at EcoPast laboratory. We used color determination to obtain information about stratigraphical changes–if they existed (see, for example [61]). In addition, we used an "objective" measurement of color to observe the distribution of red ochre in samples and as a first step to distinguish groups of samples before analyzing other soil properties. Changes in color are usually related to changes in composition and geochemical processes occurring in the soil/sediment as well as anthropogenic signals on them (e.g., [48]). Several authors have emphasized the use of quantitative CIELab color to avoid subjectivity bias for naked eye estimations (e.g., [48,61]).

Grain size analysis was performed in fine-earth samples using a set of sieves to determine the percentage of coarse sand (2–0.5 mm), medium sand (0.5–0.2 mm), fine sand (0.2–0.05 mm) and silt+clay (<0.05 mm). Percentages of each component were calculated. We used the information about grain size as a proxy for soil/sediment texture, classifying them according to the major component and observing differences between samples with and without red ochre. In addition, to explore differences in soil/sediment reaction, pH was measured in water (pHw) and KCl ($pH_{KCl}$) suspensions (ratio 1:2.5) with a pHmeter [62] using fine-earth samples. $pH_w$ indicates actual acidity/alkalinity of the soil, while $pH_{KCl}$ is related to potential acidity. ΔpH was calculated as $pH_w$—$pH_{KCl}$ and enables determining the dominant charge of the finer soil particles. Changes in soil were explored using descriptive and non-parametric statistics (Kruskal-Wallis test for independent samples, K-W test) with IBM SPSS Statistics v24. The significant value used was p<0.05.

## Identification of fatty acids

The fatty acids (FAs) were analyzed in a subsample of soil as FA methyl esters (FAMEs) by gas chromatography mass spectrometry (GC-MS). The fatty material (fat droplets) was collected from the top of a Milli-Q water layer and excess water was dried under a flow of nitrogen before adding hexane and 1% $H_2SO_4$ in methanol. The sample was transmethylated at +95˚C under nitrogen atmosphere for 110 minutes, and after cooling the solution, water and hexane were added. The formed FAMEs were recovered in hexane, which sample solution was dried with anhydrous $Na_2SO_4$, and concentrated.

The sample FAME structures were identified based on their electron impact mass spectra recorded by GCMS-QP2010 Ultra controlled by GCMSsolution V4.30 software (Shimadzu Scientific Instruments, Kyoto, Japan) and compared to published reference mass spectra [63]. The GC-MS was equipped with a Zebron ZB-wax capillary column (30 m, 0.25 mm ID and film thickness 0.25 μm; Phenomenex, Torrence CA, USA). The FAs, which were observed at a level close to the detection limit, were quantified by integrating the peak areas using the total ion current chromatograms (TIC) and, in the case of the minor saturated FAs, also using the chromatograms of their intense m/z 74 McLafferty rearrangement ion. The peak areas were

subsequently converted to mol% data by employing quantitative FAME standards. The full results are published in this work, except the FAME 22:1, reported to be an artifactual derivative of a fatty amide 22:1 originating from a plastic bag [64,65] was omitted from the FA profiles. The compositional differences between the samples were studied by Principal Component Analysis (PCA) using log-transformed mol% data (for better normality of the distribution), and the results illustrated as a heat map. These analyses were conducted by using MetaboAnalyst V5.0.

## Results

### Microparticles

**Feathers.** We identified 24 bird feather barbules, which originated mostly from the plumulaceous (downy) parts of the feathers. Also, three pennaceous barbule fragments were detected. The length of the barbules varied between 0.2–1.4 mm. Of the 21 plumulaceous barbules, seven were identified as originating from waterfowl (Anseriformes) and one tentatively from a bird of prey in the family of Falcons (Falconidae). The others shared no diagnostic features Fig 4E and S1 Table.

Feather fragments were detected from eight soil samples from layers 3, 4–5 and 6–7 so that eight fragments were found from the layers 3 and 4–5, i.e. from the same layers in which also the teeth and quartz items were recovered. The remaining 16 fragments were found from layer 6–7, in which the heavily ochre stained soil clearly started to diminish, indicating the original bottom of the grave. Horizontally, feathers were found in the middle section of the grave, and from the head area (Fig 5A).

**Animal hairs.** We detected 24 animal hairs from layers 4–5 (14 hairs) and 6–7 (10 hairs). No hairs were found from layer 3. The length of the hairs varied from 0.5 to 9.5 mm so that the average length in the bottommost layer was longer (2.88 mm) than in the uppermost layer (2.12 mm). The identification of the hairs was challenging due to their length and preservation and, therefore, most of the hairs remained unidentified Fig 5B and S1 Table.

The hairs of small rodents were identified from three samples from layers 4–5 and 6–7. It is highly probable that some other unidentified hairs were small rodent hairs, too. Although the hairs of small rodents are most probably secondary by nature, they were found only in samples taken from the original grave area Fig 4D.

From sample 868, a canid (Canidae) hair fragment was identified from layer 6–7, i.e. from the bottom of the ochre layer (Fig 4A). From the same sample, an unspecified carnivore (Carnivora) hair was identified, too (Fig 4B). Also, the hair K1 detected in sample 836 was identified as a possible canid (Fig 4C) and K4 in sample 781 as a carnivore.

**Plant fibers.** We identified three bast fibers from sample 868 in layer 6–7. Possible species in the Mesolithic context are nettle (*Urtica dioica*) and tree basts such as willow (*Salix* sp.; [2,5]), lime (*Tilia* sp.; [4]) and oak (*Quercus* sp.) (see [10] with references). Because of the poor preservation of the fibers it was impossible to make any further suggestions about the origins or species of the fibers Figs 4F and 5C and S1 Table.

**Other observations.** Besides fibers, also charred wood particles and phytoliths were detected. Phytoliths were found especially from secondary ochre samples together with modern/recent wood remains. However, it is possible that some of the phytoliths were from the grave context.

Some of the soil samples (e.g., samples 795, 796 and 836) had a greasy or oily appearance, and in water sieving small grease bubbles accumulated in the sieve. These bubbles were extracted to Eppendorf tubes to be analyzed further in the Lipidomics unit (HiLIPID) of the University of Helsinki.

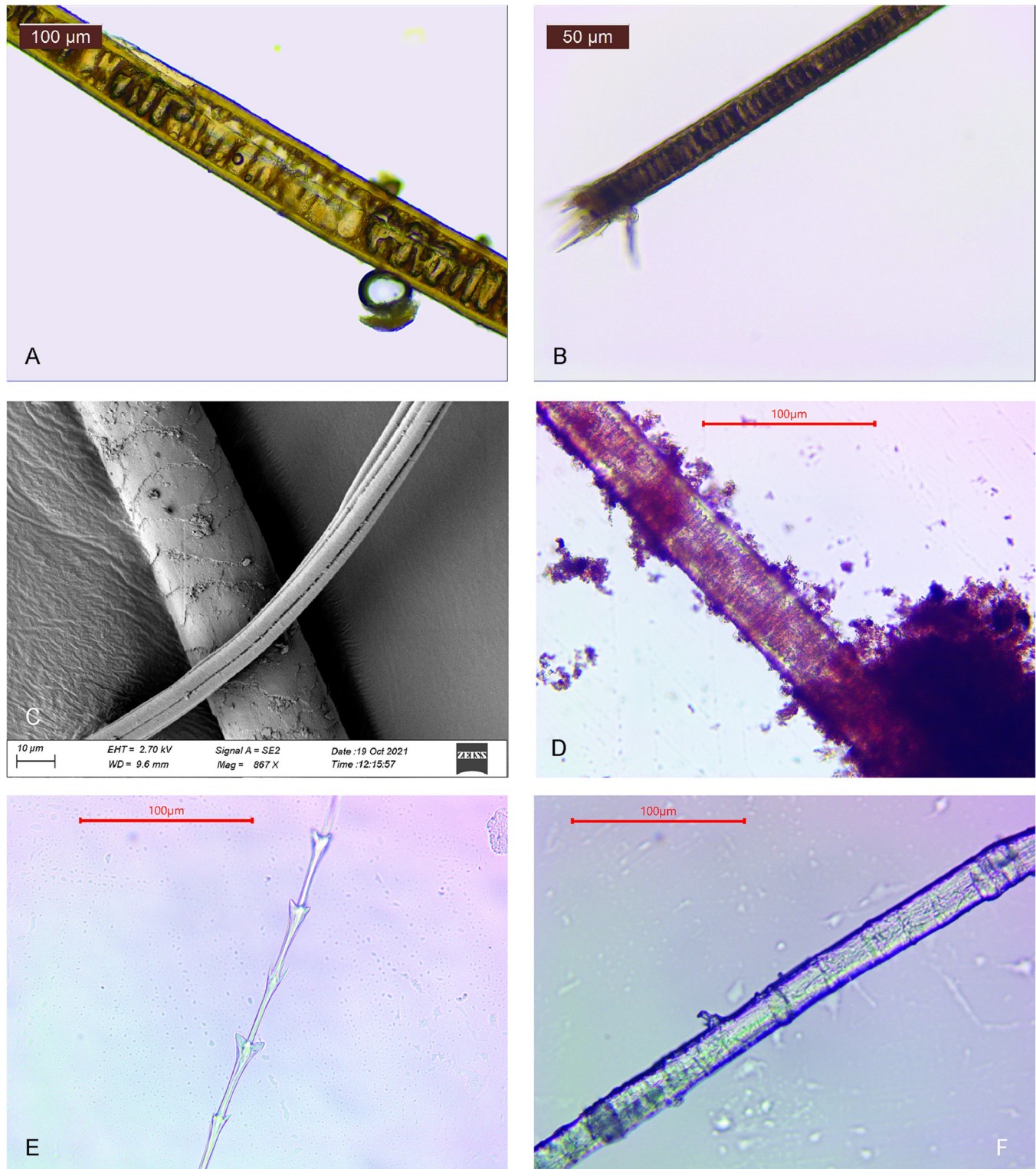

**Fig 4.** Hairs of a A) canid; B) carnivore; C) possibly canid; D) possibly European mole; E) waterfowl and E) a bast fiber. Photos: Tuija Kirkinen.

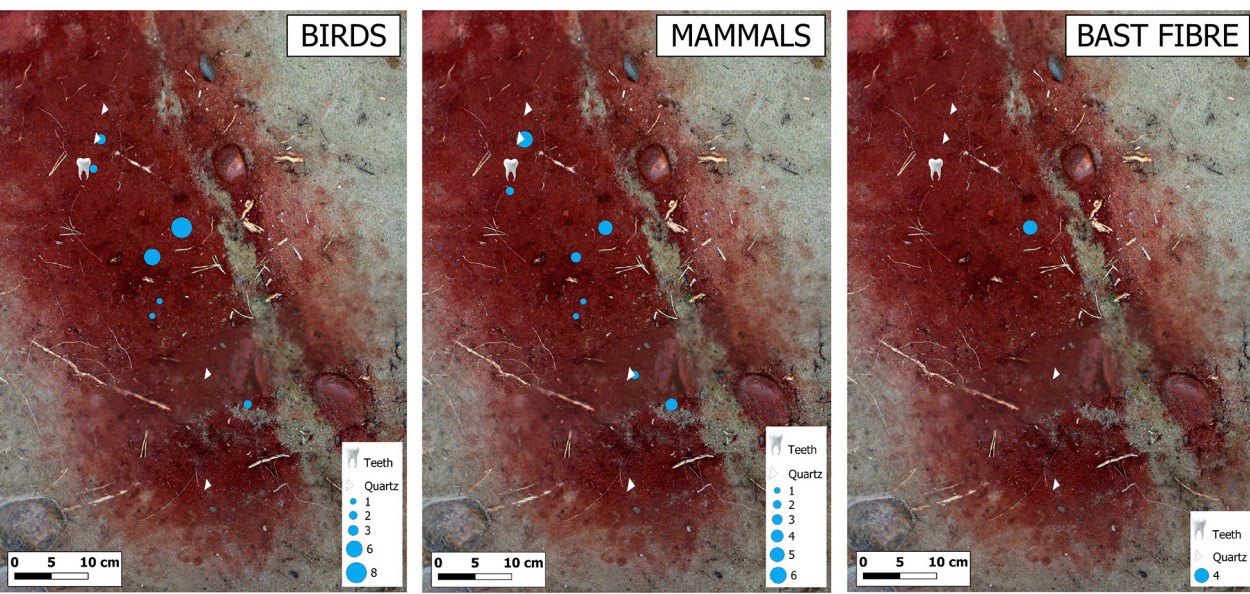

**Fig 5.** The distribution of A) bird barbules, B) mammalian hairs and C) bast fibers in the soil samples. Background map: Orthophoto of the photogrammetry 3D model: Jan-Erik Nyman, Esa Mikkola and Janne Rantanen, Finnish Heritage Agency. Drawing: Johanna Roiha.

## Soil analyses

Samples are explored regarding their color obtaining a great variability as it can be seen with the naked eye. The parameters a* and b* are correlated. We were able to identify three groups according to red and yellow intensity (Fig 6). Group-1 has been defined as those samples with an a*<3.5 and b<12 and represent samples with no ochre. A total of 20 samples are included in Group-1. Group-2 has been defined as those samples with an a* 3.5–15 and b* 12–20. A total of 14 samples falls into this group. Group-3 is comprised by 12 samples with values a*>15 and b*>20. Parent local material seems to correspond to Group-1, since its redness component is lower than in the other two groups (see Table 1 in S2 Appendix). Group-2 and -3 seem to follow the same color mixing line, departing from the average values of Group-1 (see Fig 1 in S2 Appendix) with different additions of red ochre, from low content (Group-2) to high content (Group-3). Differences among the three groups are statistically significant for all color parameters (K-W test p < 0.001; L*27.8, a*39.2; b*39.2; df = 2). See S2 Appendix and S2 Table.

The physical properties of the soil samples indicate that the samples are sandy, mainly composed of medium sands (>50%, see Table 1 in S2 Appendix). They have a minimum amount of silt and clay (~1%). Regarding the chemical properties and, as expected for the area, the samples are all acidic; overall, the charge of soils colloids (i.e., clays) is negative (ΔpH>0, n = 28), although some samples have a positive charge (ΔpH>0, n = 18)–those with higher read ochre content, in particular. The three described groups have slightly different grain size composition, as there is a significant decrease in coarse sands Group-1 > Group-2 > Group-3, having Group-3 samples a more homogeneous content of coarse sands than the other groups. A significant increase in fine sands and silt+clay follows the sequence Group1 < Group-2 < Group-3 (Table 1). The samples also have a significant decrease in pHw: Group-1 > Group-2 > Group-3, and Group-3 has a larger number of samples with negative ΔpH.

## Fatty acids

The FAs detected in the samples were mainly saturated FAs (SFAs) (See Table 1 in S3 Appendix). The two quantitatively most important FAs of all studied samples were palmitic acid

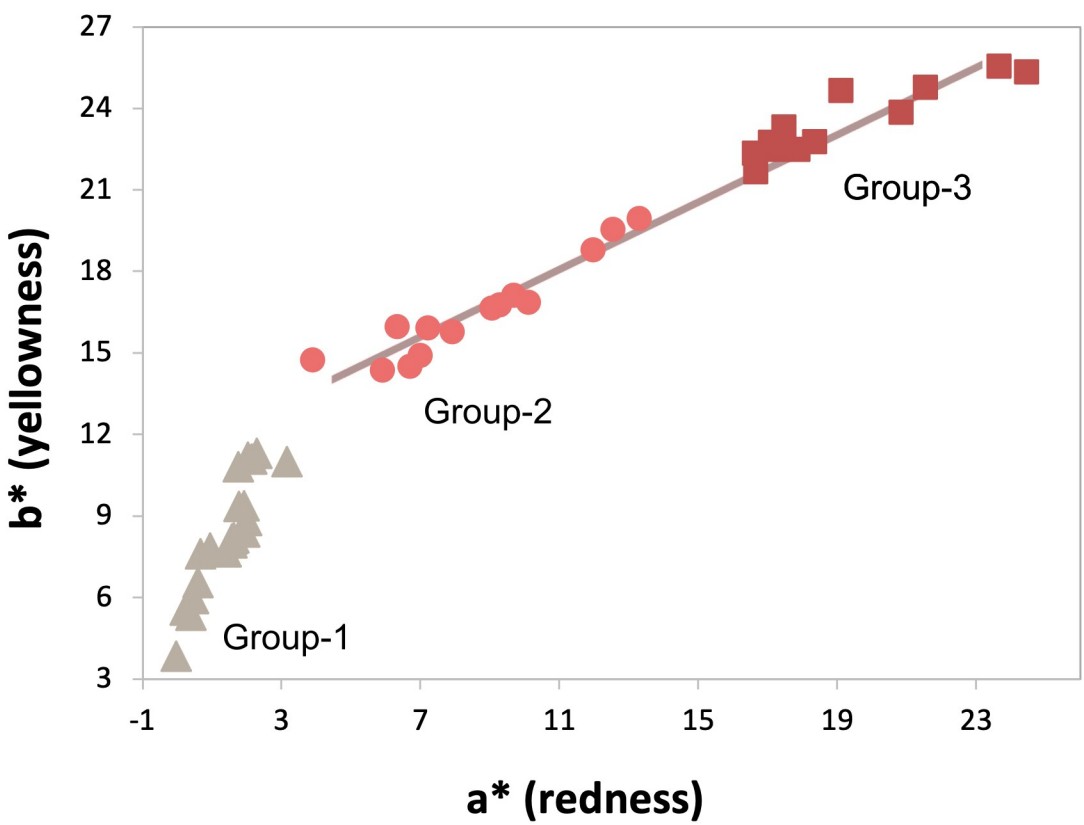

**Fig 6. Scatter plot of a\* (more/less red) and b\* (more/less yellow) parameters in all the samples analyzed, the three groups according to color parameters are identified.** Note that group-2 and -3 samples are in the same mixing line (grey line).

(16:0) and stearic acid (18:0). The greasy soil samples 795 and 836, that contained the largest number of avian barbules and mammalian hairs, were separated in PCA (PC1 describing 75%, PC2 12%, and PC3 9% of total data variation) and on the consequent heat map from the other samples (numbers 233, 450, 796 and 841) by containing larger mol% of unsaturated FAs (oleic acid 18:1n-9, and linoleic acid 18:2n-6) and long chain SFAs (20:0, 22:0 and 24:0) (See Fig 1 in S3 Appendix). In addition, the different relative concentrations of several minor FA components increased the chemical variability among the samples.

**Table 1. Descriptive statistics of the grain size and pH in the three groups described and the total of the samples.**

|  | *Total*<br>*Avg±SD (Max; min)* | *Group 1, no red*<br>*Avg±SD (Max; min)* | *Group 2, low red*<br>*Avg±SD (Max; min)* | *Group 3, red*<br>*Avg±SD (Max; min)* | *K-W test*<br>*\*significant* |
|---|---|---|---|---|---|
| % Coarse sand | *25.3±7.7 (45.6; 12.6)* | *28.6±7.7 (45.5; 15.4)* | *24.9±8.5 (40.6; 12.6)* | *20.3±3.0 (24.9; 16.3)* | *9.9, df = 2; p = 0.007\** |
| % Medium sand | *57.9±7.5 (74.6; 47.5)* | *57.0±7.9 (69.4; 48.5)* | *57.4±9.3 (74.6; 47.5)* | *60.2±3.3 (65.8; 52.3)* | *2.6 df = 2; p = 0.277* |
| % Fine sand | *15.5±4.9 (21.9;2.5)* | *13.6±6.2 (21.9; 2.5)* | *16.5±3.2 (20.3; 11.1)* | *17.6±2.3 (21.3; 14.4)* | *3.6 df = 2; p = 0.167* |
| % Silt and clay | *1.2±0.7 (2.4; 0)* | *0.9±0.7 (2.1; 0)* | *1.2±0.5 (2.0; 0.5)* | *1.9±0.4 (2.4; 1.2)* | *16.5 df = 2; p = 0.000\** |
| pH water | *4.8±0.7 (6.0; 3.9)* | *5.2±0.5 (6.0; 4.0)* | *4.6±0.5 (5.4; 3.9)* | *4.3±0.5 (5.8; 3.9)* | *16.1 df = 2; p<0.001\** |
| pH KCl | *4.5±0.2 (5.2; 3.8)* | *4.54±0.2 (5.2, 4.1)* | *4.4±0.2 (4.7; 4.1)* | *4.5±0.2 (4.7; 3.8)* | *2.0 df = 2; p = 0.366* |
| ΔpH | *0.31±0.7(1.4; -0.8)* | *0.7±0.5 (1.4; -0.8)* | *0.2±0.6 (1.25; -0.7)* | *-0.2±0.5 (1.1; -0.7)* | *13.5 df = 2; p = 0.001\** |

Kruskall-Wallis (K-W) non-parametric to tests for significant differences among the results of the three groups are also included.

## Discussion

This study is the first microscopic analysis of soil from a Mesolithic grave in Northern Europe that also combines soil properties and lipid analyses. Although it is generally considered that the organic materials have not been preserved in such graves, our study demonstrates that this is not actually always the case. So far, a few things are known about the biological profile and anthropological features of the individual buried in Majoonsuo. The human remains are compatible with a child of maximum 10.5 years of age. If red ochre covered the whole grave, the rather small size of the stained feature (~1 m) fits well with the age estimation of the deceased. The child was buried either near the contemporary settlement site, or near the one that was used much earlier. Transverse arrowheads are the only artifacts that can be interpreted as grave goods. On the other hand, in Zvejnieki, Latvia, Stone Age ochre graves have been partly filled with soils taken from the occupation layer containing artifacts [66], and we cannot exclude such as happening also in Majoonsuo. However, the find composition as well as the homogenous soil content detected in Majoonsuo soil analyses do not support this interpretation. As we cannot date the grave with radiocarbon dating, the transverse arrowheads and shore displacement support the hypothesis of Mesolithic as the most probable chronology of the grave.

### Organic soft materials in the grave

The results of microparticle analyses indicate that the grave was furnished with waterfowl down. Downy feather barbules have been detected in Mesolithic and Neolithic contexts also in Yuzhiy Oleniy Ostrov (Russia), Riņņukalns (Latvia) and Donkalnis (Lithuania) [40]. A downy layer might have been used to separate the body from the immediate contact with the sediment (see [67]. During the Late Iron Age and historical period, down has been interpreted as remains of downy filled pillows and quilts [10,39,68–71].

The single falcon barbule can be hypothesized to originate from the fletching of an arrow [72], which theory is supported by two quartz arrowheads found from the grave. However, bird feathers might also originate from wings placed in the grave or attached to garments or ornaments of head gear [40,73]. Also, we can assume that the grave was probably furnished with canid skin, or the child wore clothes made of or decorated with canid skin [74]. The canid hairs observed in the burial can derive either from a domestic dog or a wolf (*Canis lupus*). Wolf has not been identified in Finnish Stone Age bone assemblages [75], but wolf teeth are found in human burials in Latvia and in Northwestern Russia [76,77]. Dogs were present in Finland as early as humans had arrived after the last glaciation, almost 11,000 years ago [78]. Based on occasional hair fragments, it is not possible to estimate whether a complete animal or just part of the fur was buried. Fur sections or patches may have been used in clothes, shoes, and accessories. Dogs are known to have been buried together with humans, for example in Mesolithic Scandinavia [79]. Most Mesolithic animal burials involved dogs, with examples of this practice in Denmark, Netherlands, Poland, Portugal, Russia, Serbia, and Sweden; in these cases, dogs were buried alone or together with humans [43]. However, we can speculate that if a complete animal was buried in Majoonsuo, then some enamel fragments would have been preserved (just like human enamel was preserved). This points to the hypothesis that the hair fragments derive from a canid skin.

In previous research, the evidence of Mesolithic plant fibers has been detected especially from under water and wet contexts [2,4]. In Finland, Early Mesolithic net remains have been found in Antrea Korpilahti (current Russia) along wetland drainage in 1913 [80,81]. In this case, the fiber was identified as willow bast [82]. The Mesolithic finds evidence that technically,

plant fibers were looped, twined, and knotted to produce ropes, strings, and nets [83], and needle-binded to make textiles [2,5].

The wrapping or covering of bodies in fur, skins, and birch bark in ochre graves has been hypothesized by several researchers [84–86]. In general, we can also use the term *soft container* (c.f.) for tight wrapping (*effet de parois*) as well as all kinds of under covers and grave furnishings [67]. The evidence of wrapping in the Mesolithic and Neolithic Northern Europe has been ~~revealed~~ hypothesized by archaeothanatological analysis by Liv Nilsson Stutz [67] and Mari Tõrv [87]. However, we must bear in mind that our knowledge is largely based on ethnographical studies and archaeological finds from much younger periods.

The origin of the detected wood particles in the grave remains unsolved, but we can hypothesize that the body may have been wrapped in bark, the grave may have been underlaid and / or covered with bark, or the body was buried in a wooden structure like a boat. However, the latter possibility is unlikely as there was no visible evidence like dark features, which thick wooden structures should have left in the soil and soil chemical properties. Also, the presence of ochre might indicate the covering or clothing of the deceased in fur. According to some suggestions, ochre might have been used to paint the clothes (or the body itself) of the buried individual [41,66]. In the better-preserved hunter-gatherer graves of Scandinavia and the Baltic countries, the intensive use of ochre–often documented as an oval-shape feature–has been connected to soft containers [67,88]. Considering the shape and size of the Majoonsuo ochre feature, such a container might well have been used also in this burial.

## Preservation of animal fibers was detected in soil chemical analysis

The survival of hairs and feathers in Stone Age soil sediments is a subject which has been noted already in the 1990s [89–91]. During recent years, bird barbules have been detected in Mesolithic contexts in Donkalnis in west Lithuania [38], Yuzhniy Oleniy Ostrov, by Lake Onega in northwest Russia, and in Neolithic Riņņukalns in north Latvia [40]. Compared to hairs in which β-keratin dominates, feathers are composed mostly of α-keratin which is a stronger type of keratin [12]. Therefore, minuscule barbules can be assumed to have better chances of surviving in archaeological contexts.

Performing chemical analyses of the soil samples might provide supplementary information on the presence of animal material in the grave. Despite the fatty appearance of the soil samples, only trace amounts of FAs were observed, most of them being at the detection limit of sensitive GC-MS equipment. Still, the FA profiles indicated a difference between the two samples that contained the largest number of avian barbules and mammalian hairs and the rest of the samples. Despite the unsaturated and long chain FAs, present with higher percentages in those two samples, are characteristic of both mammals and birds [92,93], aerobic, acidic, and iron-rich soil is expected to have oxidized most of the unsaturated FAs, leaving SFAs behind. Thus, this study is not able to give any direct proof for connecting the unsaturated FAs to the animal material placed in the grave in prehistoric times. One potential origin of unsaturated and long chain FAs in soil are microscopic fungi, invertebrates, and microbes [94–96]. The growth and density of these soil organisms may reflect differences in the chemical composition of the soil, potentially influenced by the organic and inorganic materials buried in the sampling locations.

The soil properties studied indicate that the burial of the child was made in the local soil, without adding exogenous material to condition the burial surface. Results also suggest that a homogeneous red ochre was added, since all samples in Group-2 (low red color) and Group-3 (intense red color) are in the same mixing line (Fig 6). It is also possible that the burial area was prepared by removing the coarser material, since this fraction is lower and more

homogeneous in samples with red ochre. The addition of red ochre (iron oxides) changes the soil chemical properties by increasing its acidity, changing the dominant charge of the soil particles, and also by increasing the content of the finer particles (fine sand and silt+clay) (Table 1). A more acid environment can reduce bacterial attack on organic remains, enhancing their preservation. The charge of the silt+clay (ΔpH) suggests that soil colloids can release H+ increasing acidity and, as mentioned above, in samples with abundant red ochre, the positive charge of the colloids may be responsible for the oxidation of the unsaturated FAs. Since all samples are dominated by medium sands, the soil was probably well-drained and aerated, creating an aerobic environment that theoretically facilitates bacterial attack. However, and despite not being the most abundant fraction, local small accumulations of silt-clay can concentrate on animal fibers' surfaces covering them and protecting hairs and feathers from those bacteria that can live under such acidic conditions.

The distribution of hair finds corresponds with the primary find and ochre area of the grave. The finds, i.e., two human teeth and quartz arrow heads were detected from layer 3. Most hairs–seven fine hairs and one guard hair–were recovered from layer 4/5. Both layers are characterized by heavy concentrations of ochre. The possible canid hairs were found in sample 868, taken from layer 6/7, clearly situated at the bottom of the grave. The detection of fibers below the actual find layer is logical as small particles tend to move downwards with water, depending on the grain size of the soil. They also might originate from materials which were placed on the bottom of the grave.

The contamination of the material and bioturbation is an issue that needs to be discussed, too. First, the finding of small rodent hairs in the samples indicates the animal disturbance of burial layers. In [36], the transportation and the lining of mouse holes by fibers has been documented in an archaeological context. Also in our case, the activity of small rodents is evidenced by their hairs found in the soil samples. Studies on forensic cases indicate that scavenging behavior differs depending on the animal, but also on seasonality, rates of decomposition, and insect activity [97]. Rodent marks are the most important [98] However, it is difficult to assess if vertebrate scavengers might have caused small-scale bioturbation, without preserved bones to check the presence of bite marks, fractures and fragmentation [99] Moreover, although sandy soils are more prone to bioturbation due to the less compact structure, all analyzed soil samples share similar physical properties (any outlier was found), suggesting that the contamination, if any, was minimum. It should be stressed, however, that the rodent hairs could be from the furs and therefore part of the burial.

Second, the contamination of samples during the microparticle analyses is an issue, which is difficult to control because contaminants can be deposited in the samples already in the field, in the laboratory, and in preparing the slides. Despite cleaning the lab and analyzing room surfaces, "dust" is always in the air around us and we might even exhale small particles onto samples [100]. However, we are satisfied that our results set forth the choices made for the Mesolithic burial ritual thousands of years ago. This is evidenced by the context of the finds, as fibers were not found in the reference sample or in the samples taken outside of the grave. Also, we found no modern artificial fibers, wool or cotton other than cellulose from the cleaning equipment. Also, our test tapes from the surfaces testified to the effectiveness of the cleaning procedure.

## Conclusions

This study revealed the first animal hairs from a Finnish Mesolithic funerary context. A partly disturbed inhumation burial with a strong ochre feature was excavated in eastern Finland. Only some unburned human enamel fragments remained of the deceased. Based on these

teeth, the deceased was a child, less than 10.5 years old. Of the grave, we can say that it was made in the local soil very likely without adding exogenous materials, except homogeneous red ochre. Similar physical properties of the analyzed soil samples point to a minimal or non-existing contamination, although we cannot discard small-scale changes caused by scavengers. The grave was an inhumation and it contained feathers, canid (and potentially rodent) furs, and bast fibers used in furnishing the grave and ornaments or clothes. A falcon feather could have also been used in fletching the transverse quartz arrows found in the grave. The canids hairs might indicate the placing of a dog in the grave, but the absence of animal teeth points to the use of canid fur.

The chemical analysis could distinguish the samples with most hairs and barbules from the other samples, but due to selective oxidative degradation of unsaturated FAs [101], we could not connect these distinctive but transformed FA profiles to any specific animal. We argue that soil sampling should be applied in all burial excavations, also in areas and contexts with seemingly poor preservation of organic materials. Our results have proved that careful sampling and documenting of soils from graves, and their investigation for the discovery of micro-particles is a very promising new approach in Mortuary Archaeology. This method can be applied everywhere, but it should bring valuable information especially in contexts where organic materials have been poorly preserved. In such contexts, analyses of micro-remains and the physical and chemical properties of soil can reveal the otherwise undetectable presence of animals, and uses of furs, feathers, and plant fibers, as well as discard or evaluate possible ancient and modern contaminations or bioturbations.

## Supporting information

**S1 Table. Fibers identified from the soil samples.**
(XLSX)

**S2 Table. Soil data.**
(XLSX)

**S1 Appendix. The grave feature and soil sampling.** Background map: Orthophoto of the photogrammetry 3D model, made by Jan-Erik Nyman, Esa Mikkola and Janne Rantanen, Finnish Heritage Agency. Drawing: Johanna Roiha.
(TIF)

**S2 Appendix. Soil/sediment analysis.**
(DOCX)

**S3 Appendix. Fatty acids.**
(DOCX)

## Acknowledgments

SEM microscopy was performed on the premises of the Aalto Nanomicroscopy Center. We are grateful to Jenni Suomela for her expertise in bast fiber identification, and Krista Wright for her help with SEM facilities. We thank Sara Långsjö for her help with the research materials and Noelia Rivero Chaver for her help in the soil-analysis laboratory.

## Author Contributions

**Conceptualization:** Tuija Kirkinen, Olalla López-Costas, Antonio Martínez Cortizas, Kristiina Mannermaa.

**Data curation:** Tuija Kirkinen, Esa Mikkola.

**Formal analysis:** Olalla López-Costas, Antonio Martínez Cortizas, Reijo Käkelä.

**Funding acquisition:** Kristiina Mannermaa.

**Investigation:** Tuija Kirkinen, Olalla López-Costas, Antonio Martínez Cortizas, Sanna P. Sihvo, Hanna Ruhanen, Jan-Erik Nyman, Esa Mikkola, Janne Rantanen, Kristiina Mannermaa.

**Methodology:** Tuija Kirkinen, Hanna Ruhanen.

**Project administration:** Tuija Kirkinen.

**Resources:** Olalla López-Costas, Antonio Martínez Cortizas, Hanna Ruhanen, Reijo Käkelä.

**Visualization:** Tuija Kirkinen, Olalla López-Costas, Antonio Martínez Cortizas, Jan-Erik Nyman, Johanna Roiha.

**Writing – original draft:** Tuija Kirkinen, Olalla López-Costas, Antonio Martínez Cortizas, Sanna P. Sihvo, Hanna Ruhanen, Reijo Käkelä, Jan-Erik Nyman, Esa Mikkola, Janne Rantanen, Kristiina Mannermaa.

**Writing – review & editing:** Tuija Kirkinen, Olalla López-Costas, Antonio Martínez Cortizas, Sanna P. Sihvo, Hanna Ruhanen, Reijo Käkelä, Esa Hertell, Marja Ahola, Kristiina Mannermaa.

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
