## [Decision Letter · Decision Letter 0]

1 Aug 2022

PONE-D-22-16382Preservation of microscopic fur, feather, and bast fibers in the Mesolithic ochre grave of Majoonsuo, Eastern FinlandPLOS ONE

Dear Dr. Kirkinen,

Thank you for submitting your manuscript to PLOS ONE. After careful consideration, we feel that it has merit but does not fully meet PLOS ONE’s publication criteria as it currently stands. Therefore, we invite you to submit a revised version of the manuscript that addresses the points raised during the review process.

All comments have to addressed before re-submission.

We look forward to receiving your revised manuscript.

Kind regards,

Peter F. Biehl, PhD

Academic Editor

PLOS ONE

Journal Requirements:

3. We note that Figure 1 in your submission contain map image which may be copyrighted. All PLOS content is published under the Creative Commons Attribution License (CC BY 4.0), which means that the manuscript, images, and Supporting Information files will be freely available online, and any third party is permitted to access, download, copy, distribute, and use these materials in any way, even commercially, with proper attribution. For these reasons, we cannot publish previously copyrighted maps or satellite images created using proprietary data, such as Google software (Google Maps, Street View, and Earth). For more information, see our copyright guidelines: http://journals.plos.org/plosone/s/licenses-and-copyright.

Additional Editor Comments:

All comments have to addressed before re-submission.

Reviewers' comments:

Reviewer's Responses to Questions

**Comments to the Author**

1. Is the manuscript technically sound, and do the data support the conclusions?

Reviewer #1: Partly

2. Has the statistical analysis been performed appropriately and rigorously? 

Reviewer #1: Yes

3. Have the authors made all data underlying the findings in their manuscript fully available?

Reviewer #1: Yes

4. Is the manuscript presented in an intelligible fashion and written in standard English?

Reviewer #1: Yes

5. Review Comments to the Author

Reviewer #1: The authors have submitted a paper presenting the results of their analytical study finding evidence for bird feathers, animal hair and bast fibers in a Mesolithic child burial in Finland. The text is well structured and written in good English. The data is presented in full including tables, graphics and photographs of the site and samples taken.

The study is interesting and important for archaeological research. However, the manuscript needs some revision:

1) The conclusions are too uncritical, even if no bioturbation was "macroscopically" seen. Studies of criminal technicians on the remains of scavangers and experiments with buried animal bodies are not considered. References to these studies are missing.

2) The text includes several mistakes and incomplete archaeological comparative data:

a) The Republic of Finland is part of Northern Europe not Northeastern Europe (see lines e.g. 62-63 or 376). Moreover, in Finland but not in Northeastern Europe (e.g. NW Russia, Baltic States) human remains and unburnt animals bone are rarely recovered.

b) The authors cite the archaeothanatological analysis for the evidence for wrapping (lines 422-425), but do not mention that our knowledge is largely based on ethnographical studies and archaeological finds from much younger periods (see references).

c) The size of the Mesolithic child is compared to the size of a modern child (see Diskussion, line 382-383). However, the Mesolithic population was smaller. It would be better to refer to the children buried at the Mesolithic cemeteries, e.g. at Skateholm I, Olenii Ostrov or Zvejnieki.

d) Soft organic materials can also be preserved as imprints in soil (even from the Upper Palaeolithic period).

e) Some references need checking.

6. PLOS authors have the option to publish the peer review history of their article (what does this mean?). If published, this will include your full peer review and any attached files.

Reviewer #1: No

---

## [Author Response · Author response to Decision Letter 0]

26 Aug 2022

Response to Reviewer

/ Preservation of microscopic fur, feather, and bast fibers in the Mesolithic ochre grave of Majoonsuo, Eastern Finland

We thank the reviewer for his/her valuable comments! Here are our responses:

1 The conclusions are too uncritical, even if no bioturbation was "macroscopically" seen. Studies of criminal technicians on the remains of scavangers and experiments with buried animal bodies are not considered. References to these studies are missing.

• According to the reviewer's suggestion, the conclusions regarding the bioturbation have now been nuanced and we have added the possibility of scavengers action: “Similar physical properties of the analyzed soil samples point to a minimal or non-existing contamination, although we cannot discard small-scale changes caused by scavengers.” We have also added “very likely “ in the previous phrase. In addition, we have now included reference to forensic experimental studies on the discussion: “Studies on forensic cases indicate that scavenging behavior differs depending on the animal, but also on seasonality, rates of decomposition, and insect activity (Young et al 2014). Rodent marks are the most important (Toledo, et al 2017 ). However, it is difficult to assess if vertebrate scavengers might have caused small-scale bioturbation, without preserved bones to check the presence of bite marks, fractures and fragmentation (Young 2017)”. 

• We would like to remark that our conclusion regarding the lack of bioturbation was not only based on the lack of macroscopical evidence. In contrast, we based it on a high resolution geochemical study: the 60 analysed soil samples showed quite similar physico-chemical properties without outliers. 

2 The text includes several mistakes and incomplete archaeological comparative data:

a) The Republic of Finland is part of Northern Europe not Northeastern Europe (see lines e.g. 62-63 or 376). Moreover, in Finland but not in Northeastern Europe (e.g. NW Russia, Baltic States) human remains and unburnt animal bones are rarely recovered.

• We corrected above mentioned geographical definitions as recommended

b) The authors cite the archaeothanatological analysis for the evidence for wrapping (lines 422-425), but do not mention that our knowledge is largely based on ethnographical studies and archaeological finds from much younger periods (see references).

• First, we added a clarification that we can also use the term soft container (c.f.) for tight wrapping (effet de parois) as well as all kinds of under covers and grave furnishings (c.f. Nilsson Stutz 2003, 295–304). 

• Second, we replaced some wrapping terms with soft containers

• Finally, we added a sentence that our knowledge is largely based on ethnographical studies and archaeological finds from much younger periods as recommended

c) The size of the Mesolithic child is compared to the size of a modern child (see Diskussion, line 382-383). However, the Mesolithic population was smaller. It would be better to refer to the children buried at the Mesolithic cemeteries, e.g. at Skateholm I, Olenii Ostrov or Zvejnieki.

-The size estimation of Mesolithic children is not necessary here. We have deleted the following sentence and the reference: A current growth reference for Finland indicates a height of 1.2 m at 10-years-old for boys and girls [65]. 

d) Soft organic materials can also be preserved as imprints in soil (even from the Upper Palaeolithic period).

• That is true. We added a sentence “Even if the fibers themselves have decayed, their imprints e.g. in burnt clay carry information of hairs and plant fibers [13]”. The new reference is Tortosa et al. 2020.

e) Some references need checking.

• We have checked the references, removed one reference and added three references. 

Response to Journal Requirements

1 We have done our best to follow PLOS ONE’s style requirements. We renamed the files logically after the instructions.

2 The ethics statement of the permits required for archaeological field work and analysis was moved from Acknowledgements to the Methods section

3 The maps in Figure 1 are both CCBY 4.0 international licensed and can be used freely in the publications. We added the source information both in the maps themselves and in the figure caption. The small scale map (map A, Finland with a dot) is ESRI’s National Geographic Basemap, sources National Geographic, ESRI, Garmin, HERE, UNEP-WCMC, USGS, NASA, ESA, METI, NRCAN, GEBCO, NOAA, INCREMENT P. The large-scale map (Map B) is modified on the basis of a map of the National Land Survey of Finland, licensed under a Creative Commons Attribution 4.0 International License. More information about its copyright issues can be found in https://www.maanmittauslaitos.fi/en/opendata-licence-cc40 . By modification we refer here to shore-level reconstruction. This is mentioned in the figure text and the figure caption. The geographical information of archaeological sites is based on the Finnish Heritage Agency Register, CCBY 4.0. For more information, see https://www.museovirasto.fi/en/services-and-guidelines/data-systems/kulttuuriympaeristoen-tietojaerjestelmae/kulttuuriympaeristoen-paikkatietoaineistot

4 We have reviewed the reference list and deleted the former reference 65 because it is not necessary, and added three new references [97, 98 and 99]

Additional information

We have resubmitted Fig 1, which has been modified after reviewer comments, and S5 Appendix, in which we have corrected Table xxx with Table 1.

---

## [Editor Report · Decision Letter 1]

6 Sep 2022

Preservation of microscopic fur, feather, and bast fibers in the Mesolithic ochre grave of Majoonsuo, Eastern Finland

PONE-D-22-16382R1

Dear Dr. Kirkinen,

We’re pleased to inform you that your manuscript has been judged scientifically suitable for publication and will be formally accepted for publication once it meets all outstanding technical requirements.

Kind regards,

Peter F. Biehl, PhD

Academic Editor

PLOS ONE
---

## [Editor Report · Acceptance letter]

15 Sep 2022

PONE-D-22-16382R1 

Preservation of microscopic fur, feather, and bast fibers in the Mesolithic ochre grave of Majoonsuo, Eastern Finland 

Dear Dr. Kirkinen:

I'm pleased to inform you that your manuscript has been deemed suitable for publication in PLOS ONE. Congratulations! Your manuscript is now with our production department. 

Kind regards, 

on behalf of

Dr. Peter F. Biehl 

Academic Editor

PLOS ONE